# Green Synthesis of Iridium Nanoparticles from Winery Waste and Their Catalytic Effectiveness in Water Decontamination

**DOI:** 10.3390/ma16052060

**Published:** 2023-03-02

**Authors:** Lucia Mergola, Luigi Carbone, Tiziana Stomeo, Roberta Del Sole

**Affiliations:** 1Department of Engineering for Innovation, University of Salento, Via per Monteroni Km 1, 73100 Lecce, Italy; 2National Nanotechnology Laboratory (NNL), Institute of Nanoscience CNR c/o Campus Ecotekne, Via Monteroni, 73100 Lecce, Italy; 3Center for Bio-Molecular Nanotechnology, Istituto Italiano di Tecnologia, Via Bersanti 14, Arnesano, 73010 Lecce, Italy

**Keywords:** iridium nanoparticles, grape marc extracts, green synthesis, catalytic activity, organic dye reduction, methylene blue, TEM analysis

## Abstract

An environmentally friendly procedure was adopted for the first time to prepare green iridium nanoparticles starting from grape marc extracts. Grape marcs, waste of Negramaro winery production, were subjected to aqueous thermal extraction at different temperatures (45, 65, 80, and 100 °C) and characterized in terms of total phenolic contents, reducing sugars, and antioxidant activity. The results obtained showed an important effect of temperature with higher amounts of polyphenols and reducing sugars and antioxidant activity in the extracts with the increase of temperature. All four extracts were used as starting materials to synthesize different iridium nanoparticles (Ir-NP1, Ir-NP2, Ir-NP3, and Ir-NP4) that were characterized by Uv-Vis spectroscopy, transmission electron microscopy, and dynamic light scattering. TEM analysis revealed the presence of very small particles in all samples with sizes in the range of 3.0–4.5 nm with the presence of a second fraction of larger nanoparticles (7.5–17.0 nm) for Ir-NPs prepared with extracts obtained at higher temperatures (Ir-NP3 and Ir-NP4). Since the wastewater remediation of toxic organic contaminants on catalytic reduction has gained much attention, the application of the prepared Ir-NPs as catalysts towards the reduction of methylene blue (MB), chosen as the organic dye model, was evaluated. The efficient catalytic activity of Ir-NPs in the reduction of MB by NaBH_4_ was demonstrated and Ir-NP2 was prepared using the extract obtained at 65 °C, showing the best catalytic performance, with a rate constant of 0.527 ± 0.012 min^−1^ and MB reduction of 96.1% in just six min, with stability for over 10 months.

## 1. Introduction

Recently, wastewater remediation of toxic organic contaminants on catalytic reduction has gained great attention. Among others, organic dyes, which are widely used as colorants in several industries, are considered toxic contaminants, and dangerous for human health and aquatic life. A convenient solution for them could be to convert hazardous dyes to non-toxic chemicals and in this context, the use of noble metal nanoparticles (NPs) as todays catalysts represents a proper approach [1,2]. Herein, with the purpose to get novel catalysts for organic dye remediation, iridium NPs were prepared for the first time with a green procedure.

At present, nanoscience represents an emerging area of modern research that studies the structure of materials at a small scale (1–100 nm), evaluating their unique physical and chemical properties compared with the same bulk materials, with a high number of important applications in a wide range of sectors (electronics, packaging, engineering, cosmetics, nanomedicine, and catalysis) [3]. These important characteristics attracted the interest of the scientific community, focusing their attention on the development of new breeds of nanomaterials.

In the last years a lot of works and reviews, on the preparation of noble metal nanomaterials have been published, driven by their peculiar catalytic, optical, and electronic behavior, which allowed their pioneering use in biomedicine and engineering [4,5,6]. Among them, silver and gold NPs were the most investigated thanks to their well-known localized surface plasmon resonance that confers to the nanomaterials’ unique optical properties. Even if noble metal iridium at the nanoscale exhibits important properties such as superior stability, corrosion resistance, and catalytic activity due to the distinctive electronic configuration and the large surface area [7], it has been little investigated. These considerations move us to exploit iridium nanoparticles in the present research.

In general, noble metal NPs preparation can be carried out through physical and chemical methods such as microemulsion techniques [8], thermal decomposition [9], UV irradiation, and chemical reduction. Normally, the chemical synthesis of metal NPs needs the reduction of a metal salt by using a reducing agent in the presence of ligands, surfactants, or polymers as stabilizing agents to avoid nanoparticle agglomeration. Sodium borohydride [10], aniline [11], trisodium citrate [12], and formaldehyde [13] are the most common chemical reducing agents. Other elements such as solvent, synthesis temperature, and the type of stabilizing agent, significantly affect metal NPs characteristics. However, the use of these hazardous chemical compounds, dangerous for the environment, considerably hinders their applications, especially in nanomedicine.

Nowadays, the integration of green chemistry principles into nanotechnology is one of the key issues of nanoscience research. Indeed, there is a growing need to develop environmentally friendly and sustainable methods for the synthesis of nanoparticles that utilize nontoxic chemicals, environmentally benign solvents, and renewable materials, which are also in line with some of the fundamental principles of the circular economy [14,15,16]. In this context, recently, some reviews were published describing the possible use of vegetable extracts obtained from plants and agricultural waste as precursors for the green synthesis of metal NPs [17]. Indeed, their high content in terms of sugars, flavonoids, polyphenols, terpenoids, and saponins that can be easily extracted through an aqueous thermal extraction, make vegetable extracts able to reduce and stabilize metal nanoparticles without the use of hazardous chemical compounds, expanding considerably their fields of application [18]. For instance, noble metal nanoparticles were prepared from a lot of vegetable extracts obtained from Tectona Grandis’ seeds [19], Sphaeranthus indicus leaves [20], the pericarp of Myristica fragrans fruits [21], Nigella sativa plants [22], Malva Verticillata leaves [23], tomatoes and grapefruits [24], and many others. On the contrary, to our knowledge, there are no works on green preparation for iridium NPs starting from natural extracts.

Among many plant sources used for nanoparticle synthesis, there are grape marcs, which are the waste of wine production made of lignocellulosic material and rich in polar substances soluble in hot water with high amounts of tannins and polyphenolic compounds [25,26]. For this reason, grape marcs were also widely studied as starting material for metal nanoparticle preparation [27,28,29]. Although silver and gold nanoparticles are the most commonly studied, only recently iridium nanoparticles have attracted significant interest as selective and active catalysts for different reactions such as CO_2_ fixation, hydrogenation, and aerobic oxidation [7,30,31]. Noble metal iridium is an element of the platinum group and at the nanoscale exhibit important properties such as superior stability, corrosion resistance, and catalytic activity due to the distinctive electronic configuration and the large surface area [7]. In a recent study, Cui et al. prepared for the first time ultra-small iridium nanoparticles using tannin as a stabilizer and a common chemical reducing agent (sodium borohydride) demonstrating their ability to reduce nitroarenes [31]. With the aim to combine iridium nanoparticles research and green chemistry principles, in this study, we report for the first time the green synthesis of iridium nanoparticles (Ir-NPs) from grape marc extracts with an evaluation of some factors that affect the synthesis and the properties of these NPs. An evaluation of total polyphenol content, antioxidant activity, and reducing sugars of grape marc extracts were made. Moreover, the synthesized iridium nanoparticles were characterized by using transmission electron microscopy (TEM), dynamic light scattering (DLS), Electrophoretic Light Scattering (ELS), and Fourier transform infrared spectroscopy (FTIR) analysis.

With the aim to use the prepared NPs in the wastewater remediation field, the application of the synthesized Ir-NPs to reduce hazardous organic dyes to non-toxic chemicals was studied. In detail, the catalytic effectiveness of the synthesized Ir-NPs in the reduction of MB dye by using NaBH_4_ as a reducing agent, chosen as an organic dye model, was investigated.

## 2. Materials and Methods

### 2.1. Materials and Chemicals

Grape marc (GM) wastes from Negroamaro winery production were supplied by a local company (Cantina Vecchia Torre S.c.a.) in the Apulia region (southern Italy). Iridium trichloride (IrCl_3_), gallic acid, methylene blue (MB), 2,2′-azino-bis(3-ethylbenzothiazoline-6-sulfonic acid), diammonium salt (ABTS), Sodium Carbonate (Na_2_CO_3_) (99%), sodium borohydride (NaBH_4_), and Potassium Persulphate (K_2_S_2_O_8_) (99%) were purchased from Sigma-Aldrich (Steinheim, Germany).

Folin–Ciocalteu’s phenol reagent and Whatman 41 filter were acquired from Merck KGaA (Darmstadt, Germany). All solutions were prepared using ultrapure water obtained from a water purification system (Human Corporation, Seoul, Korea).

### 2.2. Extract Preparation

GM wastes were firstly sun-dried for six days, milled using a grinder, and finally sieved. Then, different grape marc solution extracts were prepared by adding 5 g of dried GM to 100 mL of ultrapure water. The extraction process was conducted for 1 h in a thermostat oil bath at different temperatures (45, 65, 80, and 100 °C) obtaining four extracts (GM1, GM2, GM3, and GM4, respectively)

All extracts were filtered using Whatman 41 filters. Then, centrifuged twice at 8000 rpm for 30 min and finally filtered with a 0.22 µm syringe filter. Afterward, all extracts were aliquoted and stored at −20 °C till further use. Moreover, the pH of all extracts was evaluated using a pH meter Basic 20, (Crison, Alella, Barcelona, Spain, and Europe, www.crisoninstruments.com; accessed on 27 February 2023).

### 2.3. Total Phenolic (TP) Content and Antioxidant Activity of GM Extracts

The TP content of all extracts was determined by a modified Folin–Ciocalteu colorimetric method [32] using gallic acid as standard. Briefly, 0.3 mL of each extract was mixed with Folin–Ciocalteu phenol reagent (1.5 mL) and allowed to stand to react for 6 min at room temperature. The mixtures were neutralized by 7% Na_2_CO_3_. Then, all solutions were stirred (200 rpm) for 120 min in the dark at room temperature. Phenolic compounds present in the extracts were oxidized while phosphomolybdic and phosphotungstic acid, contained in the Folin-Ciocalteu reagent, were reduced producing molybdenum (blue-colored) and tungsten oxides. After incubation, the absorbance of all samples was measured at 760 nm using a Jasco V-660 UV-visible spectrophotometer (Jasco, Palo Alto, CA, USA). The standard curves with aqueous gallic acid solutions (ranging from 30 to 600 mg/L) were used for calibration. TP was expressed as the mean of gallic acid equivalents (GAE) per gram of Dried Matter (DM) ± SD for triplicates.

The antioxidant activity of all extracts was evaluated by using an ABTS assay. This methodology is based on the ABTS cation radical formation (ABTS^•+^) generated through the oxidation of ABTS by potassium persulfate, as reported by Re et al. with slight variations [33]. Briefly, ABTS aqueous solution (7 mM) was prepared and mixed with 2.45 mM of K_2_S_2_O_8_ (final concentration). The mixture was kept in the dark for 12–15 h, under stirring (200 rpm) to favor the ABTS radical formation. After, the solution was diluted with ethanol to obtain an absorbance of 0.7 ± 0.1 at 734 nm. A calibration curve was made using known concentrations of Trolox in ethanol ranging from 0 to 15 µM. Then, 20 µL of Trolox standard solutions or sample extracts were added to 1.980 mL of ABTS^•+^ working solution, and the absorbance at 734 nm after 2.5 min of incubation, was evaluated. The scavenging capability of ABTS^•+^ radical was calculated as follows:(1)$${ABTS}^{•+}scavenging\;effect\left(\%\right)=\left[1-\left(\frac{A_{A}}{A_{B}}\right)\right]\times 100$$
where *A_B_* is the absorbance of the initial concentration of the ABTS^•+^ working solution and *A_A_* is the absorbance of the remaining concentration of ABTS^•+^ in the presence of extract. The activity of extracts was estimated at a minimum of three different concentrations. Finally, the antioxidant activity was calculated as milligrams of gallic acid (GAE) equivalents per 100 g of dried matter (DM).

### 2.4. Evaluation of Reducing Agents

Reducing sugars contained in all extracts were quantified by the Fehling method. 10 mL of Fehling solution was mixed with 3 drops of 1% blue methylene and heated. After boiling, the Fehling solution was titrated with GM extracts until a color change from blue to colorless and the formation of a red precipitate demonstrated the complete reduction of the Fehling solution. The corresponding volume of GM extract was considered for quantitative evaluation using the following equation:(2)$$\left[monosaccarides\right]\left(gL^{-1}\right)=0.05154\times 1000\times \frac{D}{A}$$
where *D* and *A* are respectively the dilution factor and the volume of GM extract used for titration.

### 2.5. Synthesis of Iridium Nanoparticles (Ir-NPs)

Synthesis of iridium nanoparticles was carried out by adding 4 mL of IrCl_3_ aqueous solution (0.005 M) and 4 mL of the grape marc extract in a glass flask. The mixture was put in a thermostatic oil bath at 80 °C under magnetic stirring (700 rpm) for 5 h. Then, all samples were centrifuged five times at 9500 rpm for 30 min using a PK121 multispeed of Thermo Electron Corporation (Thermo Electron Corporation, Waltham, MA, USA, www.thermoscientific.com; accessed on 27 February 2023). Then, the supernatant was stored at 4 °C until their use.

### 2.6. Characterization Studies

Nanoparticle formation was verified by overlapping UV-visible spectra of precursors (aqueous IrCl_3_ solution and GM extracts) and reaction product properly diluted and analyzed all at the same concentration to evaluate the disappearance of characteristic peaks of IrCl_3_ at 440 and 488 nm. To determine the main functional groups of the GM extracts and to observe the functional groups involved in the interactions with the Ir nanoparticle, all extracts and the corresponding Ir-NPs were characterized by using FTIR analysis carried out with a JASCO 660 plus infrared spectrometer. Each sample was deposited on an ATR crystal sampler and dried at 40 °C. Then, FTIR spectra were registered in the range of 4000–650 cm^−1^. DLS and ELS analysis was conducted using a Malvern Zetasizer Nano ZS 90 (Worcestershire, UK) on diluted samples to evaluate the size and Zeta Potential (ZP) of hydrate nanoparticles. An evaluation of hydrodynamic diameter was made at 25 °C by measuring the autocorrelation function at a 90° scattering angle. Low-magnification TEM analyses were conducted on a Jeol JEM-1400 electron microscope (Jeol Ltd., Akishima, Tokyo, Japan) operating at 120 kV, equipped with a CCD camera ORIUS 831 from Gatan Inc. (Pleasanton, CA, USA). TEM samples for analysis were prepared by initially depositing a few drops of the NP dispersions onto a carbon-coated copper grid and then, after almost 1 min, blotting off the sample with filter paper. The grid was made to air dry in a chemical hood. X-ray powder diffraction (XRD) measurements were carried out with a Rigaku Ultima+ model diffractometer. The X-ray generator was equipped with a copper tube operating at 40 kV and irradiating the sample with a monochromatic CuKα radiation with a wavelength of 0.154 nm. XRD spectra were acquired at room temperature over the 2θ range of 20–80°. Energy dispersive X-ray (EDX) analysis of the Ir-NPs was carried out by Phenom XL SEM microscope. EDX spectra were measured with a Silicon Drift Detector (SDD) thermoelectrically cooled (LN_2_ free) having an active area of 25 mm^2^. A voltage of 15 kV was applied in order to perform the X-rays spots analysis. The samples were prepared for EDX analysis by depositing a few drops of the NPs dispersion onto a silicon wafer and subsequently dried in an oven at 40 °C before being transferred to the microscope.

### 2.7. Catalytic Activities of Ir-NPs and Kinetic Studies

The catalytic activity of synthesized IrNPs was evaluated by studying the reduction of MB by using NaBH_4_ as a reducing agent. Firstly, aqueous solutions of MB (0.0003 M) and NaBH_4_ (0.1 M) were prepared. Then, 1 mL of NaBH_4_ was added to 9 mL of MB solution and stirred for 3 min. After, 500 µL of Ir-NPs were added to the mixture and stirred for 1 min. Then, UV-visible spectra were recorded at regular intervals of time (1.5 min) and the absorbance at 664 nm (A_t_) was monitored at reaction time *t*.

Catalytic activity was also monitored without the presence of Ir-NPs (reference) and in the presence of different extracts used for nanoparticle preparation. Kinetic studies were carried out plotting ln(A_t_/A_0_) versus reaction time (t) to evaluate the reaction rate constant (k) that was used to compare the catalytic activities of all samples. Moreover, percentages of MB reduction were also evaluated using the following equation:(3)$$MB\;reduction\left(\%\right)=\left[\frac{\left(C_{0}-C_{t}\right)}{C_{0}}\right]\times 100$$
where *C*_0_ is the initial MB concentration (mol/L) and *C_t_* is MB concentration (mol/L) at reaction time *t*. Catalytic activity studies were conducted in triplicate.

## 3. Results

### 3.1. Grape Marc Extracts Preparation and Characterization

Grape marcs and stalks represent an important solid organic waste obtained from winery production. The high add-value of this waste is due to the high contents of sugars and phytochemicals such as polyphenols and pigments with important bioactive and antioxidant properties. In this work, to preserve polyphenolic content sun drying of grape marcs was chosen. Indeed, different studies demonstrated that the TP content, obtained by drying vegetables at 20 °C, was 1.7 times higher than the polyphenolic content observed at 120 °C [34]. After drying, grape marcs were also ground to reduce particle size and increase the amount of the extracted polyphenols.

It is known that the extraction time, temperature, and nature of the extraction solvent can influence the composition in terms of phenolic compounds of the extract [35,36]. Generally, conventional polyphenol extraction methods require the use of organic solvents (methanol and ethanol) often mixed with water or acidified water to form a moderate polar medium that enhances the extraction of phenolic compounds [36,37,38]. Lapornik et al. conducted a detailed study on the influence of water on phenolic extraction from grape marcs, demonstrating as the presence of 70% of organic solvent compared to water, enhances phenolic extraction [39]. However, it was also seen that the presence of organic solvent higher than 70%, reduces TP content [40]. Starting from this important information, a green approach, using only water as extraction solvent, was preferred. After sun drying, grinding, and sieving, a thermal extraction of DM in water was conducted for 1 h at different temperatures of extraction. As can be seen in Table 1, four different temperatures were considered with the aim to obtain GM extracts differently in terms of composition and to evaluate how the nature of the extract can influence nanoparticle formation. The pH of all extracts was also determined, showing the presence of an acidic environment due to the prevalence of phenolic acids [35].

The chemical composition of grape marc extracts was extensively studied revealing the presence of polar substances with a high content of tannins and other polyphenolic compounds [37,41]. For the purpose of our work, it is essential to determine the compositions of GM extracts prepared at different extraction temperatures, in terms of both the TP content and amount of reducing sugars, due to their important action as reducing and stabilizing agents in nanoparticle formation.

As can be seen in Table 1, temperature plays a critical influence on the extracted total phenols [42]. Indeed, an enhancement of TP content with the increase in extraction temperature was observed. It can be concluded that, although the use of organic solvents could increase the TP content, the green extraction adopted using only water and the acidic environment typical of GM extracts, was sufficient to obtain TP contents comparable to the results reported using a similar procedure [39]. Moreover, the low drying temperature and the grinding step adopted in this work were fundamental to preventing the loss of polyphenols [42]. A similar trend was also observed for reducing sugars with a concentration that increases with the temperature of extraction in the range of 3.7–12.9 g/L.

The antioxidant capacity of all extracts was also evaluated by measuring the decolorization of ABTS^•+^ working solution, due to the radical cation reduction, after the addition of sample extracts. After an evaluation of the percentage of inhibition, mg of GAE equivalent per 100 g of DM were calculated. As can be seen in Table 1, the results obtained showed a similar trend of polyphenolic content with an increase in antioxidant activity of the extract, with the temperature of extraction adopted. It can be concluded that the increase in temperature of extraction determines an increase of TP and reduces the sugar content, making these extracts suitable for their use as starting material for nanoparticle preparation.

### 3.2. Synthesis and Characterization of Iridium Nanoparticles

To discriminate how the concentrations of sugars and polyphenols can affect the physicochemical properties of Ir-NPs as well as their catalytic performances, all GM extracts described previously were used as reaction media for their synthesis.

UV-visible spectra of IrCl_3_ and GM precursor solutions were overlapped with Ir-NP products after appropriate dilution to achieve the same concentration present in the reaction mixture and, for all reactions, the disappearance of characteristic peaks of IrCl_3_ at 440 and 488 nm was monitored. In particular, in Figure 1 Ir-NP1 formation was reported. As can be seen, peaks typical of IrCl_3_ completely disappear after thermolytic reduction in reaction media, indicating the complete reduction of Ir^3+^ ions into Ir(0) [43].

Table 2 summarizes the main features of the Ir-NPs developed by keeping constant the general synthesis procedure occurring at 80 °C for 5 h, however, employing as reaction environment the GM extracts obtained at different temperatures.

After purification, the effective formation of Ir-NPs was ascertained via TEM analysis (Figure 2). The left column of Figure 2 shows bright-field TEM images of all four samples; discrepancies in the extraction temperatures of GM extracts affected feebly the NP shape and more interestingly the particle size. All samples showed one main size component of NPs with a dimension in the range of 3.0–4.5 nm as displayed in Figure 2a,b, and in the insets, in the lower left quadrant of all TEM pictures. In the cases of GM extracts obtained at higher temperatures (GM3 and GM4), also another fraction of NPs was detected exposing larger sizes, as well as shapes with a clear hexagonal section (Figure 2c,d). Histograms on the right side of Figure 2 detail the above distinctly. All such outcomes evidence significant differences in the chemical nature of the GM extracts that operate by favoring diverse iridium ion complexation and then promoting dissimilar paths of NP nucleation and growth. In the examples of Ir-NP3 and Ir-NP4 samples, one may hypothesize a two-stage nucleation occurring with a certain delay time, which contributes to generating two size components; furthermore, it is likewise reasonable to envisage a greater metal ion availability. Ir-NP3 and Ir-NP4 were synthesized starting from extracts prepared at higher temperatures compared to Ir-NP1 and Ir-NP2. Table 1 provides evidence that the chemical composition of each extract is significantly dependent on the extraction temperatures. Higher temperatures sensibly increase the concentration of chemical species that either behave as complexing agents of Ir-ions or as reducing agents. The NP growth occurs as follows: the Ir ions are originally chemically complexed in a more reactive form (generally known as precursors), then, when the system reaches the condition of supersaturation (at that reaction temperature), it homogeneously nucleates forming germs of the NP. The nucleation step is followed by the growth stage, during which the unreacted (and complexed) Ir ions support the enlargement of the NP. Intuitively, the presence of a large number of reducing agents of different chemicals and reductive effectiveness prolongs the times of nucleation (multi-stage nucleation); a large amount of complexing agents makes the Ir-ions largely available for growth. The expansion of the times of the nucleation and the growth thereof, facilitates the occurrence of a multi-modal distribution of the sizes.

Hydrodynamic diameters obtained via DLS analysis confirmed the existence of an organic layer surrounding the NPs, reasonably made of tannins and polyphenols, which stabilized them within the aqueous solutions and provided for an increase in the hydrodynamic size of the particles. In addition, ZP was determined to evaluate the stability of synthesized nanoparticles (Table 2). As can be seen in Table 2, ZP negative values were obtained for all samples, indicating nanoparticle repulsion and demonstrating their stability [44]. The negative value can be due to the capping action of compounds present in the extracts [45].

The XRD patterns of Ir-NP1, Ir-NP2 (Appendix A), Ir-NP3, and Ir-NP4 do not allow to observe distinct diffraction peaks, suggesting that the nanoparticles are amorphous with iridium atoms that are located in globular nanoparticles stabilized by the organic component of the extracts [46]. Even if the information does not help us to estimate the structure of the Ir-NPs, a slight and broad diffraction peak above 40° is visible, corresponding to the (111) planes of Ir(0). To evaluate the chemical nature of the Ir-NPs, elemental EDX analysis has been performed. Figure 3 shows the EDX spectrum confirming the presence of Ir in the nanoparticles. The elemental EDX analysis has also shown KCl salt and carbon of the organic substances, whereas the presence of Silicon, SiO_2_, and Rubidium is due to the wafer substrate (Appendix A).

Ir-NPs and GM extracts were also characterized by using FTIR analysis. A comparison of FTIR spectra of grape marc extract before (GM2) and after iridium nanoparticles synthesis (Ir-NP2) is shown in Figure 4. As can be seen in Figure 4a complex and broad signals of the extracted spectrum confirms the presence of several compounds in this material ranging from polyphenols, and sugars to lignocellulosic molecules. A broad peak around 3312 and 3276 cm^−1^ is due to O-H stretching vibrations. Peaks at 2933 and 2889 cm^−1^ correspond to the C-H stretching of the olefinic chains. The peak at 1716 cm^−1^ is attributed to the carbonyl C=O stretching in ester groups. The peak at 1595 cm^−1^ corresponds to the C=C stretching of aromatic rings and together with the peaks at 1337, 1304, and 1261 cm^−1^ confirm the presence of the lignocellulosic material. Finally, peaks at 1131, 1105, and 1065 cm^−1^ are attributed to the C-O stretching of alcohols and phenols [29,47,48]. The spectrum of Ir-NP2 in Figure 4b is similar to the previous one with only some variation. It is worth noting the higher intensity of the signal of carbonyl C=O stretching in ester groups and the significant variation in the region of the C-O stretching of alcohols and phenols between 1131 and 1065 cm^−1^ where the disappearance of the signal at 1105 cm^−1^ is observed, suggesting iridium interaction with these functional groups of the grape marc extracts. FTIR spectra of the other extracts and Ir-NPs showed similar behavior to the spectra described in this paragraph.

### 3.3. Catalytic Activity

In this paragraph, the applicability of Ir-NPs as catalysts of organic dye reduction was evaluated. The reduction of MB from the water was chosen as the dye model. Although the extracts-correlated differences in terms of TP content and sugar concentrations do not significantly promote major changes in the NP morphology, important side effects on the catalytic activity of the Ir-NPs are observed instead. The catalytic performances of Ir-NPs were spectroscopically investigated by monitoring the reduction of MB aqueous solutions to leuco-MB occurring in the co-presence of sodium borohydride. The reaction was conducted in the dark avoiding air contact to prevent oxidation. It is known that MB, as chloride salt, is a cationic and primary thiazine dye that yields a stable blue water solution at room temperature. MB dye has a pKa of 3.8, which turns the diluted solution slightly acidic [49]. The UV-visible analysis of MB is very important. Generally, aqueous cationic MB shows the most intense adsorption at 664 nm associated with an MB monomer, with a shoulder peak at around 612 nm attributed to MB dimer. An additional two bands appear in the ultraviolet region with peaks at 292 and 245 nm due to the substituted benzene rings. MB, initially blue-colored in an oxidizing environment, became colorless in the presence of a reducing agent because of its reduction to the leuco-MB form and the bands in the visible region disappear. Thus, in this study, the MB reduction reaction was monitored following at 21 °C the decrease of the peak at 664 nm and for more clarity, a wide wavelength range between 400 and 800 nm was registered and shown in Figure 5, Figure 6 and Figure 7, as reported in the literature for similar works [2,50,51]. A detailed study was conducted comparing, under the same experimental conditions, the spectral behavior of the solutions containing Ir-NPs with those lacking in NPs (reference), and with those containing only the corresponding temperature-related extracts in place of NPs. Reduction of MB to leuco-MB is a well-known reversible process promoted by NaBH_4_ giving a gradual discoloration of the reaction solution and monitored by following the decrease of the peak at 664 nm but also confirmed from the return at a blue-colored solution when the final reaction solution is left in the air and the oxygen turns the leuco-MB form to the initial MB one. This behavior was observed also when the reduction is performed in the presence of Ir-NPs or the extracts. In fact, as can be seen in Appendix A the increase of the peak at 664 nm is observed when the reaction mixture, after catalytic reaction by NaBH_4_ in presence of Ir-NP2, is left at the air contact. This result confirms that the reduced form of MB is in the reaction mixture and it is again oxidated to MB by the oxygen.

UV-visible spectra were recorded at regular intervals of time (1.5 min). Firstly, the reduction of MB by NaBH_4_ in a pristine system (reference) was monitored for 25 min. As shown in Figure 5, a moderate variation of the absorbance intensity at 664 nm was found, indicating a slow and inefficient MB reduction reaction rate in the absence of a nanocatalyst (Table 3). A linear correlation between the ln(A_t_/A_0_), where A_t_ and A_0_ are the absorbance at reaction time t and 0 respectively, and the reaction time confirmed the presence of a pseudo-first-order reaction (inset of Figure 5) with a constant rate, calculated from its slope, equal to 0.011 ± 0.007 min^−1^ (Table 3). The pseudo-first-order kinetic model is consistent with the well-known kinetic behavior of MB cation reduction to leuco-MB form by a reducing agent, where it could be hypothesized a simple bimolecular reaction between the reducing molecule and MB cation [52] and it is also similar to kinetic results found in the literature if catalytic metal NPs are used [2].

Similar experiments were repeated in the presence of Ir-NPs. Figure 6 shows the metal NPs’ effectiveness in MB reduction. Indeed, a significant and fast reduction reaction happened for all four cases of Ir-NPs, with narrow discrepancies. Especially compelling the reduction process in the presence of Ir-NP2 (Figure 6b). In this last case, 96.1% of dye reduction was detected within 6 min of reaction (see also sample decolorization in Figure 6b).

Figure 7 compares the performances of all Ir-NP samples after 6 min of UV-visible irradiation: the corresponding percentages of reduction were also calculated. The dependence of the percentages of reduction and rate constants (Figure 7; Table 3) on the temperature of extraction is evident. The experimental outcomes suggest that the extracts prepared in a temperature range between 65 and 80 °C when used as reaction media for the synthesis of Ir-NPs, samples Ir-NP2 and Ir-NP3 respectively, create a chemical environment passivating the metal cores that enhance their catalytic action.

The catalytic role of the environment around the nanoparticles has been also considered. To this aim, similar experiments were reproduced employing only extracts in the process of MB reduction. It is interesting noting for any extract when compared to the reference, that also the extracts showed an increase of the reaction rate in MB reduction, though less significant than IrNPs (Table 3; Figure 8), since the presence of Ir-NPs enhances the catalytic performances of the system more than 300%, confirming a significant catalytic activity of the Ir-NPs. Moreover, the trend of the catalytic activity is similar for the extracts and for the corresponding Ir-NPs systems with a maximum observed when an extraction temperature of 65 °C is used. Even if higher extraction temperatures increased total phenolic content, in reducing sugar content and antioxidant activity of the extracts, there was no observed corresponding improvement of the catalytic activity as well in the NPs synthesis suggesting that the chemical environment changes with the extraction temperature and it also affects the catalytic activity. Moreover, the presence of Ir-NP3 and Ir-NP4 of particles with large sizes probably reduces the available surface area reflecting on their catalytic performance.

Table 4 shows a comparison between different noble metal nanoparticles. Since there are only a few works on Ir-NPs preparation and no works on the use of a green synthesis approach, we added to the table also the methodology of other noble metal nanoparticles prepared in green extracts.

Stability and recyclability are important characteristics of a good catalyst [57]. The optimum Ir catalyst synthesized in the present work, Ir-NP2, was stable for a long period of 10 months showing a stable suspension that maintains DLS results and catalytic performance. On the other side, recyclability was not measured because of the small nanoparticle size which hinders their recovery from water. Nevertheless, this preliminary study demonstrated the importance of the extraction conditions to get nanoparticles with optimal catalytic properties for dye degradation application. Further efforts should be devoted to obtaining Ir-NPs in an easily recyclable system such as magnetic bimetallic NPs or by using other supports [57].

## 4. Conclusions

A green procedure for the preparation of ultra-small Ir-NPs using nontoxic starting materials was developed. Antioxidant activity, polyphenolic and sugar contents of the extracts obtained at different extraction temperatures increased as the extraction temperature rose from 45 °C to 100 °C. TEM characterization showed as the discrepancies in the extraction temperatures of GM extracts feebly affected the NP shape and more interestingly affected the particle size. All Ir-NPs showed one main size component of nanoparticles with a dimension in the range of 3.0–4.5 nm with the presence of another fraction of NPs with larger sizes, as well as shapes with a clear hexagonal section for Ir-NP3 and Ir-NP4 prepared with GM extracted at a higher temperature. DLS analysis confirmed the presence of an organic layer surrounding the NPs providing an increase in the hydrodynamic size of the particles. Although all Ir-NPs synthesized showed an effective catalytic activity in the reduction of MB by NaBH_4_, Ir-NP2 prepared with GM2 (65 °C) evidenced better performance. Probably the composition of GM2 presents the right ratio between reducing and stabilizing agents to promote the formation of NPs with homogeneous size. This study demonstrated the importance of the extraction conditions to get nanoparticles with optimal catalytic properties. An extraction temperature of 65 °C was sufficient to obtain 4.0 nm ultra-small Ir-NP with a catalytic rate of 0.527 min^−1^ and MB reduction of 96.1% in just six min and high stability for over 10 months. The green and easy preparation procedure and the excellent catalytic performances make Ir-NPs in general and Ir-NP2 in particular optimal systems for dyes degradation application.

## Figures and Tables

**Figure 1 materials-16-02060-f001:**
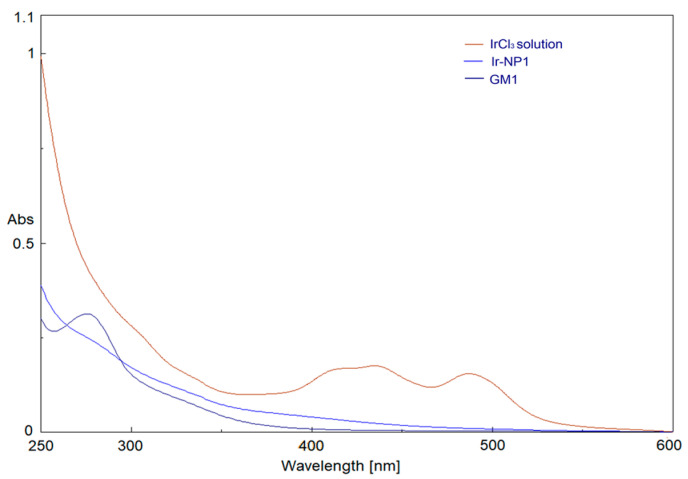
UV-Vis absorption spectra of IrCl_3_, GM1 precursors, and Ir-NP1.

**Figure 2 materials-16-02060-f002:**
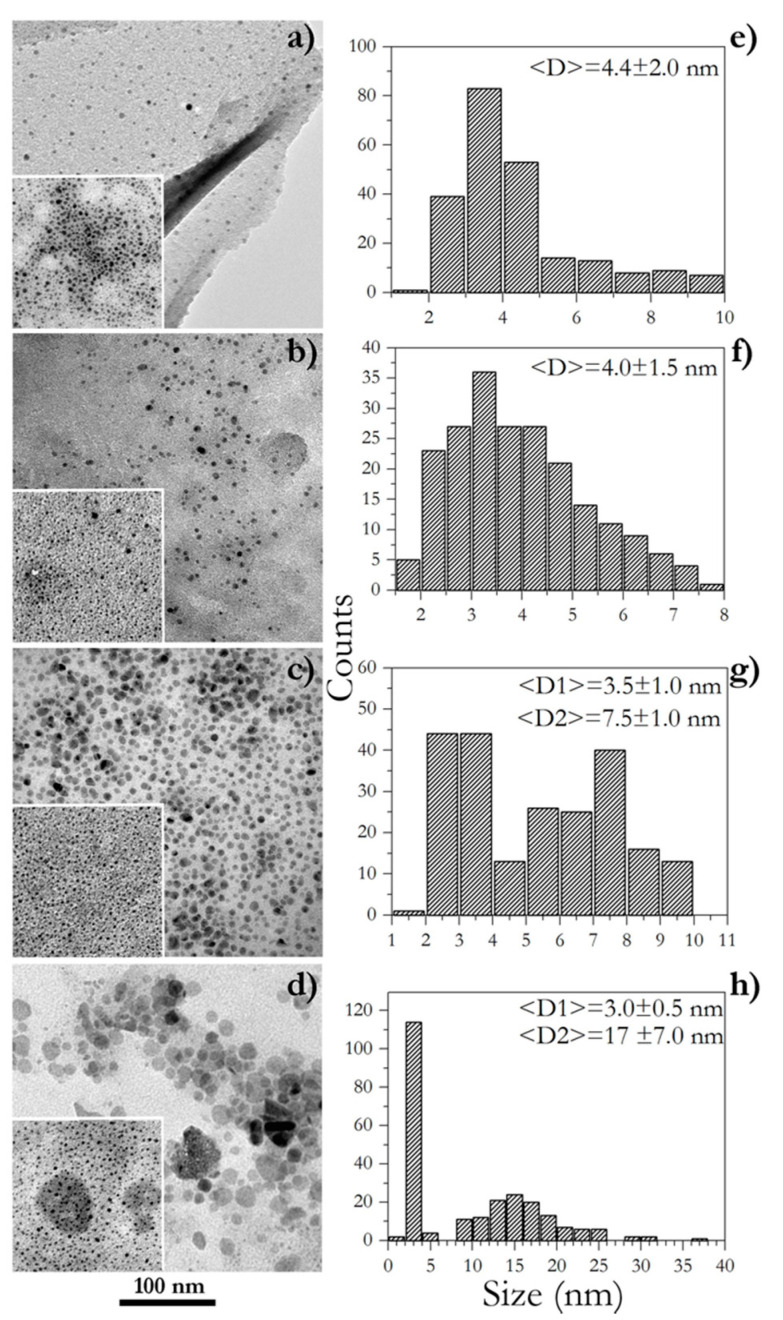
TEM analysis of Ir-NP1 (**a**), Ir-NP2 (**b**), Ir-NP3 (**c**), and Ir-NP4 (**d**), and the corresponding histograms of their size distributions (**e**–**h**).

**Figure 3 materials-16-02060-f003:**
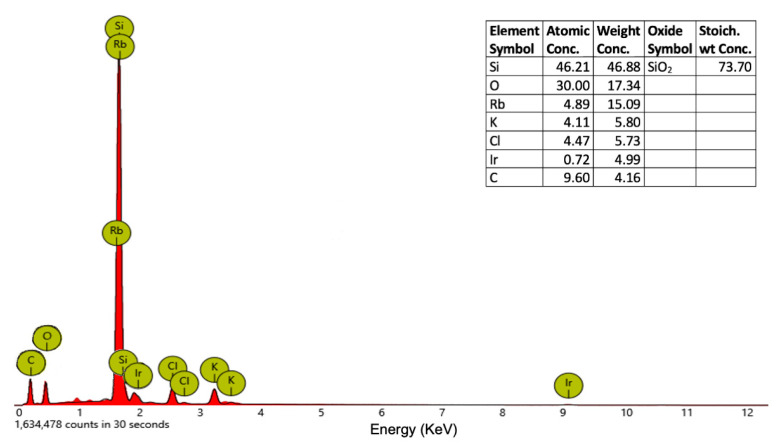
EDX spectrum and elements concentration of Ir-NP2.

**Figure 4 materials-16-02060-f004:**
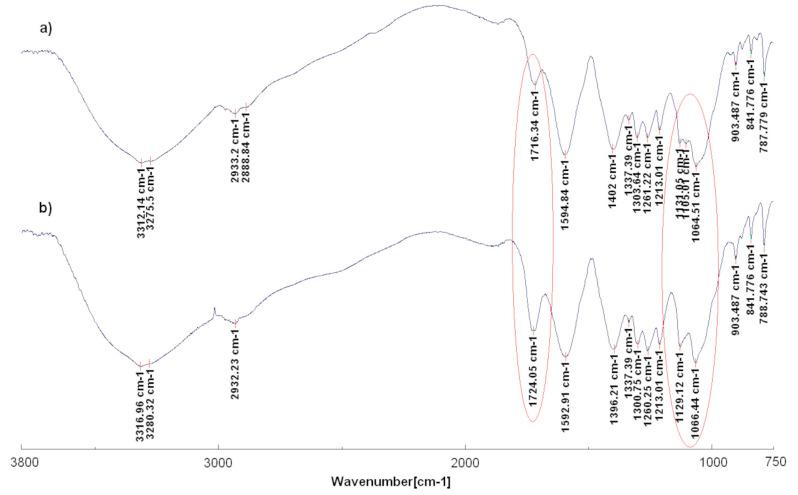
FTIR spectra of grape marc extract GM2 (**a**) and Ir-NP2 (**b**).

**Figure 5 materials-16-02060-f005:**
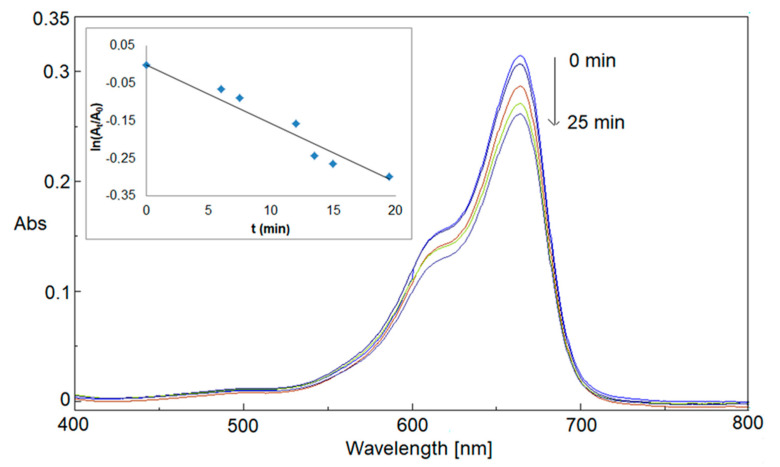
Catalytic reduction of MB by NaBH_4_ (reference).

**Figure 6 materials-16-02060-f006:**
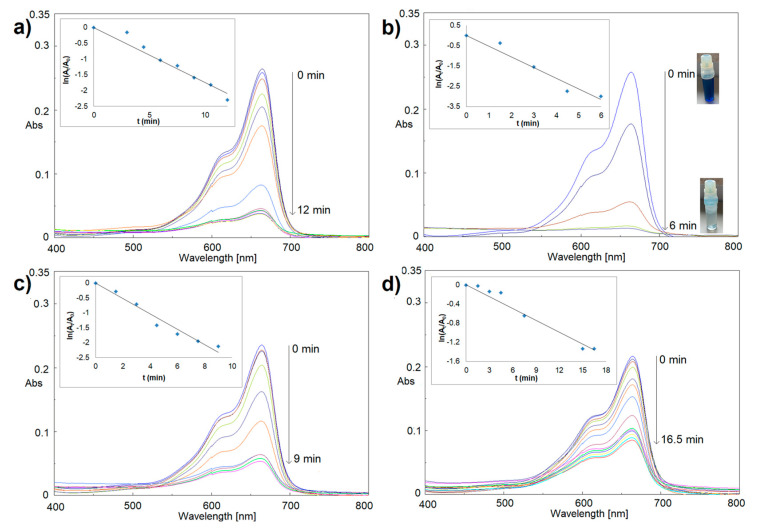
Catalytic reduction of MB by NaBH_4_ in the presence of Ir-NP1 (**a**), Ir-NP2 (**b**), Ir-NP3 (**c**), and Ir-NP4 (**d**).

**Figure 7 materials-16-02060-f007:**
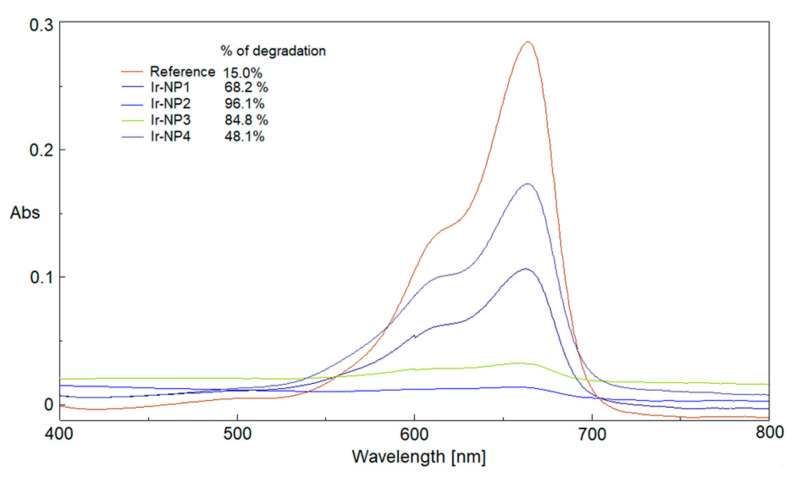
Catalytic comparison between reference and Ir-NPs at 6 min, and % of MB reduction.

**Figure 8 materials-16-02060-f008:**
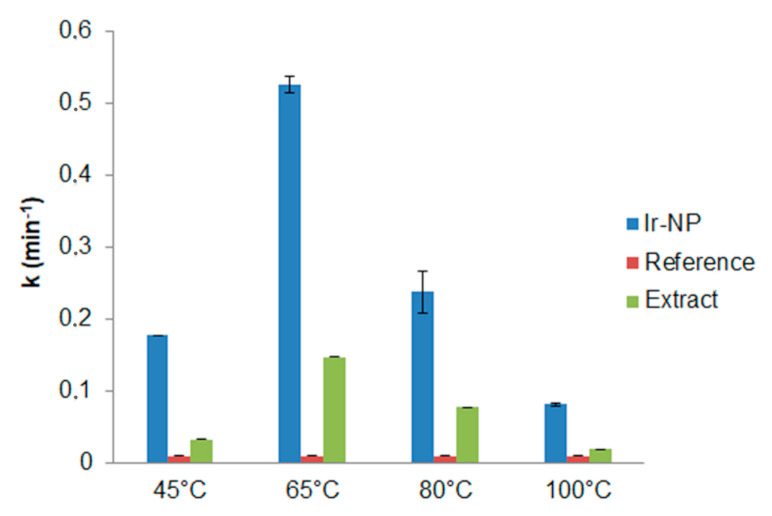
Constant rates (k) comparison between Ir-NPs, reference, and extracts.

**Table 1 materials-16-02060-t001:** Chemical composition and pH of grape marc extracts prepared at different extraction temperatures.

Extract	Temperature of Extraction (°C)	pH	TP Content (g GAE/100 g DM)	Antioxidant Activity (g GAE/100 g DM)	Reducing Sugars (g/L)
GM1	45	3.65	1.89 ± 0.10	14.9 ± 1.6	3.7 ± 0.9
GM2	65	3.50	2.85 ± 0.11	22.6 ± 1.7	6.9 ± 0.2
GM3	80	3.50	4.41 ± 0.15	38.0 ± 2.5	10.3 ± 0.7
GM4	100	3.60	6.35 ± 0.13	46.9 ± 1.9	12.9 ± 0.8

**Table 2 materials-16-02060-t002:** TEM and DLS features of synthesized Ir-NPs.

Nanoparticles Dispersion	Grape Marc Extract	TEM Size (nm)	Hydrodynamic Diameter (nm)	Zeta Potential (mV)
Ir-NP1	GM1	4.4 ± 2.0	214 ± 110	−18.6 ± 0.2
Ir-NP2	GM2	4.0 ± 1.5	184 ± 64	−14.3 ± 1.4
Ir-NP3	GM3	3.5 ± 1.0 7.5 ± 1.0	177 ± 56	−5.8 ± 0.8
Ir-NP4	GM4	3.0 ± 0.5 17.0 ± 7.0	193 ± 56	−7.6 ± 0.6

**Table 3 materials-16-02060-t003:** The specific rate constant of catalytic reduction of MB by NaBH_4_ for reference, in the presence of extracts (GM1, GM2, GM3, and GM4), and the presence of iridium nanoparticles (Ir-NP1, Ir-NP2, Ir-NP3, and Ir-NP4). Experiments were conducted in triplicate.

Sample	k (min ^−1^) ± SD	R^2^
Reference	0.011 ± 0.007	0.940
GM1	0.033 ± 0.002	0.979
GM2	0.148 ± 0.038	0.954
GM3	0.078 ± 0.012	0.965
GM4	0.020 ± 0.002	0.940
Ir-NP1	0.174 ± 0.002	0.954
Ir-NP2	0.527 ± 0.012	0.953
Ir-NP3	0.238 ± 0.029	0.966
Ir-NP4	0.082 ± 0.002	0.964

**Table 4 materials-16-02060-t004:** Comparison of catalytic results for MB degradation with literary works.

Metal NPs	Reducing Agent	Stabilizer	Size (nm)	Application	Literature
Ir-NPs	Ethylene glycol	PVP	3	-	[53]
Ir-NPs	NaBH_4_	Ascorbic acid	3.8	-	[54]
Ir/WO3	NaBH_4_	-	About 200	MB catalytic degradation	[55]
Ir-NPs	NaBH_4_	Tannic acid	3.5	Detection of H_2_O_2_ and xanthine	[30]
Ir-NPs	NaBH_4_	trisodium citrate	2.5	Detection of Sudan red I	[56]
Ir-NPs	GM extracts	GM extracts	3–4.5	MB reduction	This work
Au-NPs	seed, skin, and stalk grapes extracts	Extracts of seed, skin, and stalk of grapes	20–25	-	[28]
Ag-NPs	Grape stalk extracts	Grape stalk extracts	27.7	Electrochemical determination of Pb(II) and Cd(II)	[29]
Fe-NPs	Eucalyptus leaves extract	Eucalyptus leaves extract	40–60	Removal of Acid black 194	[46]
Ag-NPs	Grape and tomato juices	Grape and tomato juices	10–30	antibacterial and protein kinase inhibition activity	[24]
Ir-NPs	NaBH_4_	Tannic acid	3.5	Nitroarenes reduction	[31]

## Data Availability

The data presented in this study are available on request from the corresponding author.

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
