# Peer review of "Green Synthesis of Iridium Nanoparticles from Winery Waste and Their Catalytic Effectiveness in Water Decontamination"

_materials, 2023, doi:10.3390/ma16052060_

Round 1
Reviewer 1 Report
Review Materials-2200935
The manuscript reported the preparation of Ir nanoparticles (Ir NPs) from wastewater originating from winery industry. The Ir NPs were used to reduce a model dye (methylene blue). The topic discussed can be of interest of Materials reader. The following suggestions were offered to the author for the improvement of the manuscript:
1. The pH of the grape marc extract should be lower for increased extraction temperature. Was there a volume lost?
2. Why there was no two-stage nucleation occur for Ir-NP1 and Ir-NP2? Were there any other reasons for the larger Ir NPs observed for Ir-NP3 and Ir-NP4?
3. Suggest including XRD spectra (or alternatively XPS spectra) on the manuscript to understand the chemical states of the Ir NPs.
4. Suggest re-plotting the FTIR spectra – the annotation is illegible.
5. Suggest re-plotting Fig. 5. The insets are illegible.
6. Suggest explaining the result of MB reduction with GM and Ir-NPs. Why there is no trend observed for the reduction. Were the experiments conducted in triplicate?
7. Comment on the stability of MB in different pH condition.
8. Suggest comparing the results obtained with other recently published paper.
Author Response
REPLY TO REVIEWER 1
We wish to thank you for the evaluation and comments about our manuscript.
We have modified the manuscript taking into account your comments/suggestions.
All changes have been written in red in the manuscript.
Please find below our answers to your comments:
Point 1. The pH of the grape marc extract should be lower for increased extraction temperature. Was there a volume lost?
Response 1. Actually, we carefully prepared the experimental set up for the extraction process by using a refrigerant to avoid solvent loss, thus we are confident that the volume did not change. Probably, it is not easy to correlate pH value with the temperature extraction since the composition of the extract is complicated and there are different factors that can affect the pH environment.
Point 2. Why there was no two-stage nucleation occur for Ir-NP1 and Ir-NP2? Were there any other reasons for the larger Ir NPs observed for Ir-NP3 and Ir-NP4?
Response 2. We included the following explanation in the text following the principles of theory of the classical nucleation and growth in solution (dx.doi.org/10.1021/cr400544s, Chem. Rev. 2014, 114, 7610−7630 and references therein).
Page 8: “Ir-NP3 and Ir-NP4 were synthesized starting from extracts prepared at higher temperatures compared to Ir-NP1 and Ir-NP2. Table 1 evidences that the chemical composition of each extract is significantly dependent on the extraction temperatures. Higher temperatures sensibly increase the concentration of chemical species that either behave as complexing agents of Ir-ions or as reducing agents. The NP growth occurs as follows: the Ir ions are originally chemically complexed in a more reactive form (generally known as precursors), then, when the system reaches the condition of supersaturation (at that reaction temperature), it homogeneously nucleates forming germs of the NP. The nucleation step is followed by the growth stage, during which the unreacted (and complexed) Ir ions support the enlargement of the NP. Intuitively, the presence of a large number of reducing agents of different chemical and reductive effectiveness, prolongs the times of nucleation (multi-stage nucleation); the large amount of complexing agents makes the Ir-ions largely available for the growth. The expansion of the times of the nucleation and the growth thereof, facilitates the occurrence of a multi-modal distribution of the sizes.”
Point 3. Suggest including XRD spectra (or alternatively XPS spectra) on the manuscript to understand the chemical states of the Ir NPs.
Response 3. In supplementary material we added XRD spectrum of Ir-NP2 (Figure S1) even if the information does not help us to estimate the structure of the Ir NPs, in fact only a slight and broad diffraction peak above 40° is visible, corresponding to the (111) planes of Ir(0). Moreover we used EDX analysis to confirm the Ir presence in the nanoparticles. EDX results were reported in a new Figure 3 and in supplementary material EDX of the substrate was added (Figure S2). As a consequence all the old figures from 3 to 7 changed in the new version from 4 to 8.
Moreover, the following sentences were added into the text:
Page 5: “Energy dispersive X-ray (EDX) analysis of the Ir-NPs was carried out by Phenom XL SEM microscope. EDX spectra were measured with a Silicon Drift Detector (SDD) thermoelectrically cooled (LN2 free) having an active area of 25 mm2. A voltage of 15 kV was applied in order to perform the X-rays spots analysis. The samples were prepared for EDX analysis by depositing a few drops of the NPs dispersion onto a silicon wafer, and subsequently dried in an oven at 40°C before being transferred to the microscope.”
Page 9:”Even if the information does not help us to estimate the structure of the Ir-NPs, a slight and broad diffraction peak above 40° is visible, corresponding to the (111) planes of Ir(0). To evaluate the chemical nature of the Ir-NPs, elemental EDX analysis has been performed. Figure 3 shows the EDX spectrum confirming the presence of Ir in the nanoparticles. The elemental EDX analysis has also shown KCl salt and carbon of the organic substances, whereas the presence of Silicon, SiO2 and Rubidium is due to the wafer substrate (Figure S2).”
Point 4. Suggest re-plotting the FTIR spectra – the annotation is illegible.
Response 4. Following the proper reviewer suggestion, we increased characters size in FTIR spectra and we changed the old Figure 3 with the Figure 4 new. Page 10:
Point 5. Suggest re-plotting Fig. 5. The insets are illegible.
Response 5. As suggested from the reviewer we re-plotted the old figure 5 and upload figure 6 new as follow at page 12:
Point 6. Suggest explaining the result of MB reduction with GM and Ir-NPs. Why there is no trend observed for the reduction. Were the experiments conducted in triplicate?
Response 6. Following the reviewer suggestion we improved the explanation, adding the following sentences in Results and Discussion:
Page 13: “The catalytic role of the environment around the nanoparticles has been also considered. To this aim, similar experiments were reproduced employing the only extracts in the process of MB reduction. It is interesting noting for any extract, when compared to the reference, that also the extracts showed an increase of the reaction rate in MB reduction, though less significant than IrNPs (Table 3; Fig. 8), since the presence of Ir-NPs enhances the catalytic performances of the system more than 300%, confirming a significant catalytic activity of the Ir-NPs. Moreover, the trend of the catalytic activity is similar for the extracts and for the corresponding Ir-NPs systems with a maximum observed when an extraction temperature of 65 °C is used. Even if higher extraction temperatures increased total phenolic content, reducing sugar content and antioxidant activity of the extracts there was not observed a corresponding improvement of the catalytic activity as well in the NPs synthesis suggesting that the chemical environment changes with the extraction temperature and it also affects the catalytic activity. Moreover, the presence in Ir-NP3 and Ir-NP4 of particles with large size probably reduces the available surface area reflecting on their catalytic performance.”
Yes, the experiments were conducted in triplicate, thus the following sentences were added in the text:
Page 5: “Catalytic activity studies were conducted in triplicate” and in table 3 and figure 7 “Experiments were conducted in triplicate”
Point 7. Comment on the stability of MB in different pH condition.
Response 7. In the reducing experiments we worked at slightly acidic pH since only small volumes of the IrNPs or of the extracts were added (0.5 mL in 10.5 mL, 5%) where MB is stable as cationic MB (chloride salt) with the positive charge on the nitrogen element. To better explain this aspect some MB properties such as UV explanation, pH condition and redox reaction were added into the text:
Page 11: “The reaction was conducted in the dark avoiding air contact to prevent oxidation. It is known that MB, as chloride salt, is a cationic and primary thiazine dye that yields a stable blue water solution at room temperature. MB dye has a pKa of 3.8 which turns the diluted solution slightly acidic [49]. The UV-visible analysis of MB is very important. Generally, aqueous cationic MB shows the most intense adsorption at 664 nm associated with an MB monomer, with a shoulder peak at around 612 nm attributed to MB dimer. An additional two bands appear in the ultraviolet region with peaks at 292 and 245 nm due to the substituted benzene rings.”
and also a reference was added:
- Khan, I.; Saeed, K.; Zekker, I.; Zhang, B.; Hendi, A.H.; Ahmad, A.; Ahmad, S.; Zada, N,; Ahmad, H.; Ali Shah, L.; Shah. T.; Khan I. Methylene Blue: Its Properties, Uses, Toxicity and Photodegradation. Water 2022, 14, 242. https://doi.org/10.3390/w14020242
Point 8 Suggest comparing the results obtained with other recently published paper.
Response 8. As suggested from the reviewer we added in the text a table containing a comparison between different works. Since there are only few works on Ir-NPs preparation and no works on the use of a green synthesis approach, we added in the table also th e methodology of other noble metal nanoparticles prepared by green extracts. Thus, the following sentences were added into the text:
Page 14: “Table 4 shows a comparison between different noble metal nanoparticles. Since there are only few works on Ir-NPs preparation and no works on the use of a green synthesis approach, we added in the table also the methodology of other noble metal nanoparticles prepared in green extracts.”
Moreover, the following table 4 and some new references 51-54 were also added into the text:
Table 4. Comparison of catalytic results for MB degradation with literature works
|
Metal NPs |
Reducing agent |
Stabilizer |
Size (nm) |
Application |
Literature |
|
Ir-NPs |
Ethylene glycol |
PVP |
3 |
- |
[51] |
|
Ir-NPs |
NaBH4 |
Ascorbic acid |
3.8 |
- |
[52] |
|
Ir/WO3 |
NaBH4 |
- |
About 200 |
MB catalytic degradation |
[53] |
|
Ir-NPs |
NaBH4 |
Tannic acid |
3.5 |
Detection of H2O2 and xanthine |
[30] |
|
Ir-NPs |
NaBH4 |
trisodium citrate |
2.5 |
Detection of Sudan red I |
[54] |
|
Ir-NPs |
GM extracts |
GM extracts |
3 – 4.5 |
MB reduction |
This work |
|
Au-NPs |
seed, skin, and stalk grapes extracts |
Extracts of seed, skin, and stalk of grapes |
20-25 |
- |
[28] |
|
Ag-NPs |
Grape stalk extracts |
Grape stalk extracts |
27.7 |
Electrochemical determination of Pb(II) and Cd(II) |
[29] |
|
Fe-NPs |
Eucalyptus leaves extract |
Eucalyptus leaves extract |
40-60 |
Removal of Acid black 194 |
[46] |
|
Ag-NPs |
Grape and tomato juices |
Grape and tomato juices |
10-30 |
antibacterial and protein kinase inhibition activity |
[24] |
|
Ir-NPs |
NaBH4 |
Tannic acid |
3.5 |
Nitroarenes reduction |
[31] |
The following references were also added in references section:
- Bonet, F.; Delmas, V.; Grugeon, S.; Herrera Urbina R.; Silvert, P-Y. Synthesis of monodisperse Au, Pt, Pd, Ru and Ir nanoparticles in ethylene glycol. NanoStruct. Mat. 1999, 11(8), 1277-1284. https://doi.org/10.1016/S0965-9773(99)00419-5.
- Chakrapani, K.; Sampath, S. Spontaneous assembly of iridium nanochain-like structures: surface enhanced Raman scattering activity using visible light. Chem. Commun. 2014, 50, 3061-3063. https://doi.org/10.1039/C3CC49690B.
- Dhanalakshmi, M.; Lakshmi Prabavathi, S.; Saravanakumar, K.; Filip Jones, B.; Muthuraj, V. Iridium nanoparticles anchored WO3 nanocubes as an efficient photocatalyst for removal of refractory contaminants (crystal violet and methylene blue). Chem.Phys. Lett. 2020, 745, 137285. https://doi.org/10.1016/j.cplett.2020.137285.
- Cui, M.; Zhao, Y.; Wang, C.; Song, Q. Synthesis of 2.5 nm colloidal iridium nanoparticles with strong surface enhanced Raman scattering activity. Microchim. Acta 2016, 183, 2047-2053. https://doi.org/10.1007/s00604-016-1846-z.
I look forward to hearing from you soon
Best Regards
February 15 2023
Dr. Roberta Del Sole
Corresponding Author:
Dr. Roberta Del Sole
Department of Engineering for Innovation - University of Salento
Via per Monteroni, Km 1, 73100 Lecce , Italy
phone. +39 832 297 256; e-mail: roberta.delsole@unisalento.it

Reviewer 2 Report
Report:
The article “Green synthesis of iridium nanoparticles from winery waste and their application as catalysts of dyes reduction for water decontamination” is written well and I will recommend this manuscript after major corrections/revision.
Points to be explained
1- The author should rearrange title.
2- The abstract section looks good but authors must add comprehensive results of analysis.
3- The introduction section missing research gap. The purpose and novelty of the research should be added in the Introduction section.
4- In this context, recently, some reviews were published describing the possible use of vegetable extracts obtained from plants and agricultural waste as precursors for the green synthesis of metal NPs [15-19] ?? No need of bulk references.
5- Provide only relevant and recent literature including plant name and nanoparticles types.
6- Total phenolic content and antioxidant activities of GM extract have been tested. Did you test the heavy metal contents of GM extract.
7- The catalytic activity of synthesized IrNPs was evaluated by studying the reduction of MB from NaBH4.? Correct the sentence.
8- An insufficient characterization technique is provided. The author should add EDX and XPS of prepared nanoparticles to verify the formation of metal nanoparticles.
9- The authors did not measure the zeta potential of the as-made Ir nanoparticles.
10- The author must provide XRD pattern of Ir-NPs.
11- How author claim that the produced nnaoparticles are metallic Iridium not Iridium oxide?
12- In the catalytic activity, the author states that “MB becomes colorless in the presence of a reducing agent because of its reduction to the leuco-MB form”. How you can explain the formation of leuco form in your case? From which evidence you suggest?
13- Ir-NPs have demonstrated catalytic activity comparable to that extract GM2. This show that the reduction of Ir nanoparticles did not completed using extract only.
14- The author must describe and provide in detail about the UV-visible spectra of MB reduction. No detail is provided. Why author used the spectra from 400 to 800 nm.
15- How the authors claim that this reaction of MB by Ir-NPs is reduction not adsorption?
16- The author should measure the recyclability study of prepared nanoparticles for MB dyes.
17- The author must provide the mechanism of the synthesis of Ir-NPs from Ir3+
18- The reduction ability of Ir-NP2 is high as compared to other. Justify? Higher temperature leads to large number of nucleation of nanoparticles with higher reduction of metal nanoparticles
19- The authors claim that reduction follow pseudo-first- order reaction. Justify.
20- There are few minor grammatical, spelling mistakes to be corrected.
21- Conclusion section is very long focus should be on the findings of the research results regarding the subject of the study and how to benefit from it.
Author Response
REPLY TO REVIEWER 2
We wish to thank you for the evaluation and comments about our manuscript.
We have modified the manuscript taking into account your comments/suggestions.
All changes have been written in red in the manuscript.
Please find below our answers to your comments:
Point 1. The author should rearrange title.
Response 1. As suggested from the reviewer the title was rearranged as follow:
“Green synthesis of iridium nanoparticles from winery waste and their catalytic effectiveness in water decontamination”
Point 2. The abstract section looks good but authors must add comprehensive results of analysis.
Response 2. We thank the reviewer for the suggestion. To better explain the results of analysis we modified the abstract as follow:
“An environmentally friendly procedure was adopted for the first time to prepare green iridium nanoparticles starting from grape marc extracts. Grape marcs, waste of Negramaro winery production, were subjected to aqueous thermal extraction at different temperatures (45, 65, 80 and 100 °C) and characterized in terms of total phenolic contents, reducing sugars and antioxidant activity. The results obtained showed an important effect of temperature with higher amounts of polyphenols and reducing sugars and antioxidant activity in the extracts with the increase of temperature. All four extracts were used as starting materials to synthesize different iridium nanoparticles (Ir-NP1, Ir-NP2, Ir-NP3 and Ir-NP4) that were characterized by Uv-Vis spectroscopy, transmission electron microscopy and dynamic light scattering. TEM analysis revealed the presence of very small particles in all samples with sizes in the range of 3.0-4.5 nm with the presence of a second fraction of larger nanoparticles (7.5-17.0 nm) for Ir-NPs prepared with extracts obtained at higher temperatures (Ir-NP3 and Ir-NP4). Since the wastewater remediation of toxic organic contaminants on catalytic reduction has gained much attention, the application of the prepared Ir-NPs as catalysts towards the reduction of methylene blue (MB), chosen as organic dye model, was evaluated. The efficient catalytic activity of Ir-NPs in the reduction of MB by NaBH4 was demonstrated and Ir-NP2, prepared using the extract obtained at 65 °C, showed the best catalytic performance with a rate constant of 0.527 ± 0.012 min-1 and MB reduction of 96.1% in just six minutes with a stability for over 10 months. “
Point 3. The introduction section missing research gap. The purpose and novelty of the research should be added in the Introduction section.
Response3. We thank the reviewer for the suggestion. To better explain the purpose and novelty of the research we moved the speech about wastewater remediation from the end to the beginning of the introduction including other considerations as follow:
“Recently, wastewater remediation of toxic organic contaminants on catalytic reduction has gained great attention. Among others, organic dyes which are widespread used as colourants in several industries are considered toxic contaminants, dangerous for human health and aquatic life. A convenient solution for them could be to convert hazardous dyes to non-toxic chemicals and in this contest the use of noble metal nanoparticles (NPs) as catalysts represents today a proper approach [1,2]. Herein, with the purpose to get novel catalysts for organic dye remediation, iridium NPs were prepared for the first time with a green procedure.”
Moreover, we rearranged the final sentences of the introduction as follow:
“With the aim to use the prepared NPs in the wastewater remediation field, the application of the synthesized Ir-NPs to reduce hazardous organic dyes to non-toxic chemicals was studied…”
Point 4. In this context, recently, some reviews were published describing the possible use of vegetable extracts obtained from plants and agricultural waste as precursors for the green synthesis of metal NPs [15-19] ?? No need of bulk references.
Response 4. As properly noted from the reviewer there are many references in this context, thus we selected the most relevant one removing the references 15, 16, 18, and 19.
page 2:”..., recently, some reviews were published describing the possible use of vegetable extracts obtained from plants and agricultural waste as precursors for the green synthesis of metal NPs [17].”
Point 5. Provide only relevant and recent literature including plant name and nanoparticles types.
Response 5. As suggested from the reviewer we added in the text some examples of nanoparticles prepared using different plant source as follow:
Page 3:”For instance, noble metal nanoparticles were prepared from a lot of vegetable extracts obtained from Tectona Grandis’ seeds [19], Sphaeranthus indicus leafs [20], pericarp of Myristica fragans fruits [21], Nigella sativa plants [22], Malva Verticillata leaves [23], tomato and grapefruits [24] and many others.”
References added:
- Rautela, A.; Rani, J.; Debnath (Das), M. Green synthesis of silver nanoparticles from tectona grandis seeds extract: characterization and mechanism of antimicrobial action on different microorganisms. Anal. Sci. Technol. 2019, 10, 5. https://doi.org/10.1186/s40543-018-0163-z.
- Balalakshmi, C.; Gopinath, K.; Govindarajanc, M.; Lokesh, R.; Arumugam, A.; Alharbi, N. S.; Kadaikunnan, S.; Khaled, J. M.; Benelli, G. Green synthesis of gold nanoparticles using a cheap sphaeranthus indicus extract: impact on plant cells and the aquatic crustacean artemia nauplii. Chem. Rev. 2019, 387, 450–462. https://doi.org/10.1016/j.jphotobiol.2017.06.040.
- Sasidharan, D.; Namitha, T. R.; Johnson, S. P.; Jose, V.; Mathew, P. Synthesis of silver and copper oxide nanoparticles using myristica fragrans fruit extract: antimicrobial and catalytic applications. Chem. Pharm. 2020, 16, 100255. https://doi.org/10.1016/j.scp.2020.100255.
- Alkhalaf, M. I.; Hussein, R. H.; Hamza, A. Green synthesis of silver nanoparticles by nigella sativa extract alleviates diabetic neuropathy through anti-inflammatory and antioxidant effects. Saudi J.Biol. Sci. 2020, 27 (9), 2410-2419. https://doi.org/10.1016/j.sjbs.2020.05.005.
- Sk, I.; Khan, M. A.; Haque, A.; Ghosh, S.; Roy, D.; Homechuadhuri, S.; Alam, A. Synthesis of gold and silver nanoparticles using malva verticillata leaves extract: study of gold nanoparticles catalysed reduction of nitro-schiff bases and antibacterial activities of silver nanoparticles. Res. Green Sustainable Chem. 2020, 3, 100006. https://doi.org/10.1016/j.crgsc.2020.05.003.
- Zia, M.; Gul, S.; Akhtar, J.; Haq, I. U.; Abbasi, B. H.; Hussain, A.; Naz, S.; Chaudhary, M. F. Green synthesis of silver nanoparticles from grape and tomato juices and evaluation of biological activities. IET Nanobiotechnol. 2017, 11(2), 193-199. https://doi.org/10.1049/iet-nbt.2015.0099.
Point 6.Total phenolic content and antioxidant activities of GM extract have been tested. Did you test the heavy metal contents of GM extract
Response 6. Actually, we focused only on the extract composition that can have a higher influence on the reduction process for the formation of metal nobel nanoparticles, foregoing a complete chemical characterization of the extracts that we consider out of the aim of this work.
Point 7.The catalytic activity of synthesized IrNPs was evaluated by studying the reduction of MB from NaBH4.? Correct the sentence.
Response 7. As correctly observed from the reviewer, the above sentence is wrong, thus we modified the sentence as follow:
Page 3, at the end of the introduction section: “...In detail, the catalytic effectiveness of the synthesized Ir-NPs in the reduction of MB dye by using NaBH4 as a reducing agent, chosen as an organic dye model, was investigated.”
Point 8. An insufficient characterization technique is provided. The author should add EDX and XPS of prepared nanoparticles to verify the formation of metal nanoparticles.
Response 8. Unfortunately we did not provide XPS results because the instrument is not available. However, as suggested from the reviewer 1, to better characterize metal nanoparticles, Figure 3 with EDX results was added in the manuscript. Figure S2 was also added in supplementary material. Moreover the following sentences were added into the text:
Page 5: “Energy dispersive X-ray (EDX) analysis of the Ir-NPs was carried out by Phenom XL SEM microscope. EDX spectra were measured with a Silicon Drift Detector (SDD) thermoelectrically cooled (LN2 free) having an active area of 25 mm2. A voltage of 15 kV was applied in order to perform the X-rays spots analysis. The samples were prepared for EDX analysis by depositing a few drops of the NPs dispersion onto a silicon wafer, and subsequently dried in an oven at 40° before being transferred to the microscope.”
Page 9:”Even if the information does not help us to estimate the structure of the Ir-NPs, a slight and broad diffraction peak above 40° is visible, corresponding to the (111) planes of Ir(0). To evaluate the chemical nature of the Ir-NPs, elemental EDX analysis has been performed. Figure 3 shows the EDX spectrum confirming the presence of Ir in the nanoparticles. The elemental EDX analysis has also shown KCl salt and carbon of the organic substances, whereas the presence of Silicon, SiO2 and Rubidium is due to the wafer substrate (Figure S2).”
Point 9. The authors did not measure the zeta potential of the as-made Ir nanoparticles.
Response 9. As properly noted from the reviewer Zeta potential measurements were not reported. For this reason we evaluated also the Zeta-potential adding in table 2 this information and also a comment in the text as follow:
Page 8:“In addition, ZP was determined to evaluate the stability of synthesized nanoparticles (Table 2). As it can be seen in table 2, ZP negative values were obtained for all samples, indicating nanoparticles repulsion and demonstrating their stability [44]. The negative value can be due to the capping action of compounds present in the extracts [45].”
References 44 and 45 are added:
- Yazdi, M.; Yousefvand, A.; Hosseini, H.A.; Mirhosseini, S.A. Green synthesis of silver nanoparticles using nisin and its antibacterial activity against pseudomonas aeruginosa. Biomed. Res. 2022, 11, 56. https://doi.org/10.4103%2Fabr.abr_99_21.
- Raja, S.; Ramesh, V.; Thivaharan, V. Green biosynthesis of silver nanoparticles using Calliandra haematocephala leaf extract, their antibacterial activity and hydrogen peroxide sensing capability. J. Chem. 2017, 10, 253-261. https://doi.org/10.1016/j.arabjc.2015.06.023.
Point 10. The author must provide XRD pattern of Ir-NPs.
Response10. In supplementary material we added XRD spectrum (Figure S1) even if the information does not help us to estimate the lattice structure of the Ir NPs, in fact only a slight and broad diffraction peak above 40° is visible, corresponding to the (111) planes of Ir(0). Moreover we used EDX analysis to confirm the Ir presence. The following sentences were added into the text:
Page 9: “The XRD patterns of Ir-NP1, Ir-NP2 (Figure S1), Ir-NP3 and Ir-NP4 do not allow to observe distinct diffraction peaks, suggesting that the nanoparticles are amorphous with iridium atoms that are located in globular nanoparticles stabilized by the organic component of the extracts [46]. Even if the information does not help us to estimate the structure of the Ir-NPs, a slight and broad diffraction peak above 40° is visible, corresponding to the (111) planes of Ir(0). To evaluate the chemical nature of the Ir-NPs, elemental EDX analysis has been performed.”
Point 11. How author claim that the produced nanoparticles are metallic Iridium not Iridium oxide?
Response 11. As reported in figure 1 we used UV technique to characterize metallic iridium nanoparticles formation as reported in other works. As it can be seen in the figure the peaks characteristic of Ir3+disappear after nanoparticles formation. In order to further support this statement we added EDX of metal nanoparticles.
Point 12. In the catalytic activity, the author states that “MB becomes colorless in the presence of a reducing agent because of its reduction to the leuco-MB form”. How you can explain the formation of leuco form in your case? From which evidence you suggest?
Response 12. As correctly observed from the reviewer we did not explain enough the redox MB process that occurs during catalytic studies by using NaBH4 without or in presence of Ir NPs or extracts. Thus, we discuss the entire process in results and discussion and we also added other experimental details that are useful to justify the leucoMB formation. The following sentences were added into the text:
Page 11: “The reaction was conducted in the dark avoiding air contact to prevent oxidation. It is known that MB, as chloride salt, is a cationic and primary thiazine dye that yields a stable blue water solution at room temperature. MB dye has a pKa of 3.8 which turns the diluted solution slightly acidic [49]. The UV-visible analysis of MB is very important. Generally, aqueous cationic MB shows the most intense adsorption at 664 nm associated with an MB monomer, with a shoulder peak at around 612 nm attributed to MB dimer. An additional two bands appear in the ultraviolet region with peaks at 292 and 245 nm due to the substituted benzene rings. MB, initially blue-coloured in an oxidizing environment, became colourless in the presence of a reducing agent because of its reduction to the leuco-MB form and the bands in the visible region disappear. The MB reduction reaction was monitored in a wavelength range of 400 and 800 nm at 21 °C. A detailed study was conducted comparing, under the same experimental conditions, the spectral behaviour of the solutions containing Ir-NPs with those lacking in NPs (reference), and with those containing only the corresponding temperature-related extracts in place of NPs. Reduction of MB to leuco-MB is a well-known reversible process promoted by NaBH4 giving a gradual discoloration of the reaction solution and monitored by following the decrease of the peak at 664 nm, but also confirmed from the return at a blue-coloured solution when the final reaction solution is left at the air and the oxygen turns the leuco-MB form to the initial MB one. This behaviour was observed also when the reduction is performed in presence also of Ir-NPs or the extracts.”
Point 13. Ir-NPs have demonstrated catalytic activity comparable to that extract GM2. This show that the reduction of Ir nanoparticles did not completed using extract only.
Response 13. Unfortunately we did not explain clearly the catalytic results of the studied systems. These probably led to a misunderstanding of the results. We observed an increasing catalytic activity when Ir-NP2 is added instead of GM2 if we compare specific rate constants of catalytic reduction of MB (Table 3) which is 0.527±0.012 for Ir-NP2 and 0.148±0.038 for GM2. We think that each extract needs to be compared with the corresponding NPs synthesized. We tried to better explain the catalytic results of the studied systems into the text:
Pages 13-14: “The catalytic role of the environment around the nanoparticles has been also considered. To this aim, similar experiments were reproduced employing the only extracts in the process of MB reduction. It is interesting noting for any extract, when compared to the reference, that also the extracts showed an increase of the reaction rate in MB reduction, though less significant than IrNPs (Table 3; Fig. 8), since the presence of Ir-NPs enhances the catalytic performances of the system more than 300%, confirming a significant catalytic activity of the Ir-NPs. Moreover, the trend of the catalytic activity is similar for the extracts and for the corresponding Ir-NPs systems with a maximum observed when an extraction temperature of 65 °C is used. Even if higher extraction temperatures increased total phenolic content, reducing sugar content and antioxidant activity of the extracts there was not observed a corresponding improvement of the catalytic activity as well in the NPs synthesis suggesting that the chemical environment changes with the extraction temperature and it also affects the catalytic activity. Moreover, the presence in Ir-NP3 and Ir-NP4 of particles with large size probably reduces the available surface area reflecting on their catalytic performance.”
Point 14. The author must describe and provide in detail about the UV-visible spectra of MB reduction. No detail is provided. Why author used the spectra from 400 to 800 nm.
Response 14. Generally the reduction of MB to leucoMB is spectroscopically monitored by following the decrease of its main peak at 664 nm, thus also in our studies the main aspect was to register the signal at 664 nm. Moreover, the choice to register the spectra between 400 and 800 nm was due to the necessity to limit the time of the analysis in order to register the spectra of each analysis every 1.5 minutes. We added more details about MB UV-visible spectrum into the text:
Page 11:”The UV-visible analysis of MB is very important. Generally, aqueous cationic MB shows the most intense adsorption at 664 nm associated with an MB monomer, with a shoulder peak at around 612 nm attributed to MB dimer. An additional two bands appear in the ultraviolet region with peaks at 292 and 245 nm due to the substituted benzene rings. MB, initially blue-coloured in an oxidizing environment, became colourless in the presence of a reducing agent because of its reduction to the leuco-MB form and the bands in the visible region disappear. The MB reduction reaction was monitored in a wavelength range of 400 and 800 nm at 21 °C.”
Point 15. How the authors claim that this reaction of MB by Ir-NPs is reduction not adsorption?
Response 15. The reduction of MB is confirmed from the behaviour observed in all catalysis experiments. A reversible oxidation process has been noted at the end of reduction process if the reaction mixture is left in contact with air and the solution turns to blue colour due to oxidation of leuco-MB form to MB. We explained these observation into the text, adding the following phrase:
Page 11: “Reduction of MB to leuco-MB is a well-known reversible process promoted by NaBH4 giving a gradual discoloration of the reaction solution and monitored by following the decrease of the peak at 664 nm, but also confirmed from the return at a blue-coloured solution when the final reaction solution is left at the air and the oxygen turns the leuco-MB form to the initial MB one. This behaviour was observed also when the reduction is performed in presence also of Ir-NPs or the extracts.”
Point 16. The author should measure the recyclability study of prepared nanoparticles for MB dyes.
Response 16. This work is a preliminary study on the preparation of Ir-NP starting from natural products; for this reason unfortunately, reusability studies were not performed because of the small dimension of nanoparticles which hinders their separation. However, in the future it could be possible to create magnetic bimetallic Ir-NPs, in order to easily remove the catalyst from water. Alternatively, to better characterize Ir-NPs, stability studies were performed and better expressed into the text adding the following sentence:
Pag 15: “Stability is an important characteristic of a good catalyst. The optimum Ir catalyst synthesised in the present work, Ir-NP2, was stable for a long period of 10 months showing a stable suspension that maintains DLS results and catalytic performance.”
Point 17. The author must provide the mechanism of the synthesis of Ir-NPs from Ir3+
Response17. Actually we did not use a classical mechanism study for the synthesis of Ir-NPs such as kinetic study, intermediate formation and so on. Since the extraction solution used is complex we have chosen to evaluate some important components that it is well known have reduction properties and can participate in the reduction of Ir3+. In particular polyphenols, reducing sugars and antioxidant activity. We used the obtained results to promote some hypotheses on the correlation between them and the morphological data of the nanoparticles and on the catalytic behaviour. The previous discussions have been also deepened during this revision step. We don’t feel adding further and more detailed mechanism theories.
Point 18. The reduction ability of Ir-NP2 is high as compared to other. Justify? Higher temperature leads to large number of nucleation of nanoparticles with higher reduction of metal nanoparticles
Response 18. As partially discussed previously at point 13 we better explain the catalytic results of the studied systems into the text as follow:
Pages 13-14: “confirming a significant catalytic activity of the Ir-NPs. Moreover, the trend of the catalytic activity is similar for the extracts and for the corresponding Ir-NPs systems with a maximum observed when an extraction temperature of 65 °C is used. Even if higher extraction temperatures increased total phenolic content, reducing sugar content and antioxidant activity of the extracts there was not observed a corresponding improvement of the catalytic activity as well in the NPs synthesis suggesting that the chemical environment changes with the extraction temperature and it also affects the catalytic activity. Moreover, the presence in Ir-NP3 and Ir-NP4 of particles with large size probably reduces the available surface area reflecting on their catalytic performance.”
Point 19. The authors claim that reduction follow pseudo-first- order reaction. Justify.
Response 19. To satisfy the proper request to justify the kinetic results we added the following sentences in results and discussion:
Page 11:" The pseudo-first-order kinetic model is consistent with the well-known kinetic behaviour of MB cation reduction to leuco-MB form by a reducing agent, where it could be hypothesised a simple bimolecular reaction between the reducing molecule and MB cation [50] and it is also similar to kinetic results found in literature if catalytic metal NPs are used [2].”
Point 20. There are few minor grammatical, spelling mistakes to be corrected.
Response 20. We read the text carefully and corrected some grammatical errors.
Point 21. Conclusion section is very long focus should be on the findings of the research results regarding the subject of the study and how to benefit from it.
Response 21. Correctly the reviewer noted a too long conclusion section, thus we removed some sentences and slightly changed some phrases to reduce this paragraph as follow:
Page 15: “A green procedure for the preparation of ultra-small Ir-NPs using nontoxic starting materials was developed. Antioxidant activity, polyphenolic and sugars contents of the extracts obtained at different extraction temperatures increased as the extraction temperature rose from 45 °C till 100 °C. TEM characterization showed as the discrepancies in the extraction temperatures of GM extracts feebly affected the NP shape and more interestingly affected the particle size. All Ir-NPs showed one main size component of nanoparticles with a dimension in the range of 3.0 – 4.5 nm with the presence of another fraction of NPs with larger sizes, as well as shapes with a clear hexagonal section for Ir-NP3 and Ir-NP4 prepared with GM extracted at higher temperature. DLS analysis confirmed the presence of an organic layer surrounding the NPs providing an increase in the hydrodynamic size of the particles. Although all Ir-NPs synthesized showed an effective catalytic activity in the reduction of MB by NaBH4, Ir-NP2 prepared with GM2 (65 °C) evidenced better performance. Probably the composition of GM2 presents the right ratio between reducing and stabilizing agents to promote the formation of NPs with homogeneous size. This study demonstrated the importance of the extraction conditions to get nanoparticles with optimal catalytic properties. An extraction temperature of 65 °C was sufficient to obtain 4.0 nm ultra-small Ir-NP with a catalytic rate of 0.527 min-1 and MB reduction of 96.1% in just six minutes and high stability for over 10 months. The green and easy preparation procedure and the excellent catalytic performances make Ir-NPs in general and Ir-NP2 in particular optimal systems for dyes degradation application.”
I look forward to hearing from you soon
Best Regards
February 15 2023
Dr. Roberta Del Sole
Corresponding Author:
Dr. Roberta Del Sole
Department of Engineering for Innovation - University of Salento
Via per Monteroni, Km 1, 73100 Lecce , Italy
phone. +39 832 297 256; e-mail: roberta.delsole@unisalento.it

Reviewer 3 Report
The authors present a procedure to prepare iridium nanoparticles from grape pomace extracts in the manuscript. Grape pomace was subjected to aqueous thermal extraction at different temperatures (45, 65, 80, and 100 °C) and characterized for total phenolic content, reducing sugars, and antioxidant activity and was used as starting materials to synthesize different nanoparticles of iridium. The samples were characterized by UV-Vis spectroscopy, transmission electron microscopy, and dynamic light scattering. TEM analysis showed a particle size range between 3.0-4.5 nm with the presence of a second fraction of larger nanoparticles for Ir-NPs prepared with extracts obtained at higher temperatures. The application of Ir-NPs prepared as catalysts for MB reduction showed good catalytic performance.
The authors present a good motivation for the research, involving environmental problems of the other methodologies and an important application of the material in catalysis for water decontamination.
The methodology was well divided, which facilitates understanding.
In table 1 the authors use 3 significant figures in the uncertainty of antioxidant activity. My suggestion is to reduce the number of significant figures in the uncertainty to one or two, as more figures have little statistical significance. Remember that the measurement must follow the same number of decimal places as the measurement. The same must be followed for the other tables.
It would be interesting for the authors to improve the resolution of figure 3 and also increase the font of the texts.
Furthermore, the work is very well presented, has significant results, and should be accepted.
Author Response
REPLY TO REVIEWER 3
We wish to thank you for the evaluation and comments about our manuscript.
We have modified the manuscript taking into account your comments/suggestions.
All changes have been written in red in the manuscript.
Furthermore, the work is very well presented, has significant results, and should be accepted.
Please find below our answers to your comments:
Point 1. In table 1 the authors use 3 significant figures in the uncertainty of antioxidant activity. My suggestion is to reduce the number of significant figures in the uncertainty to one or two, as more figures have little statistical significance. Remember that the measurement must follow the same number of decimal places as the measurement. The same must be followed for the other tables.
Response 1. Correctly the reviewer observed a wrong use of significant figures. Following his suggestion we reduced the significant figures where necessary. Thus, in Table 1, Table 2 and Table 3 we changed the numbers and also where they were mentioned into the text.
Point 2. It would be interesting for the authors to improve the resolution of figure 3 and also increase the font of the texts.
Response 2. As suggested from the reviewer we improve the resolution of figure 3, also increasing the font of the texts as follow:
I look forward to hearing from you soon
Best Regards
February 15 2023
Dr. Roberta Del Sole
Corresponding Author:
Dr. Roberta Del Sole
Department of Engineering for Innovation - University of Salento
Via per Monteroni, Km 1, 73100 Lecce , Italy
phone. +39 832 297 256; e-mail: roberta.delsole@unisalento.it

Round 2
Reviewer 2 Report
The author did not explain the following points.
1- The author must describe and provide in detail about the UV-visible spectra of MB reduction. No detail is provided. Why author used the spectra from 400 to 800 nm.
2- How the authors claim that this reaction of MB by Ir-NPs is reduction not adsorption?
3- The author should measure the recyclability study of prepared nanoparticles for MB dyes.
Compare your results of catalytic reduction with DOI: 10.1002/aoc.4971
Author Response
We wish to thank you for the evaluation and comments about our manuscript.
We have modified the manuscript taking into account your comments/suggestions.
All changes have been written in red and highlighted in yellow in the manuscript.
Please find below our answers to your comments:
Point 1. The author must describe and provide in detail about the UV-visible spectra of MB reduction. No detail is provided. Why author used the spectra from 400 to 800 nm.
Response 1. In addition to the previous explanation, we added more details into the text and other references as follow (highlighted in yellow):
Page 11, “The UV-visible analysis of MB is very important. Generally, aqueous cationic MB shows the most intense adsorption at 664 nm associated with an MB monomer, with a shoulder peak at around 612 nm attributed to MB dimer. An additional two bands appear in the ultraviolet region with peaks at 292 and 245 nm due to the substituted benzene rings. MB, initially blue-colored in an oxidizing environment, became colorless in the presence of a reducing agent because of its reduction to the leuco-MB form and the bands in the visible region disappear. Thus, in this study the MB reduction reaction was monitored following at 21 °C the decrease of the peak at 664 nm and for more clarity a wide wavelength range between 400 and 800 nm was registered and shown in figures 5-7 as reported in literature for similar works [2,50,51].”
- Suvith, V.S.; Philiph, D. Catalytic degradation of methylene blue using biosynthesized gold and silver nanoparticles. Acta A Mol. Biomol. Spectrosc. 2014, 118, 526-532. http://dx.doi.org/10.1016/j.saa.2013.09.016.
- Vidhu, A.A.; Philiph, D. Spectroscopic, microscopic and catalytic properties of silver nanoparticles synthesized using Saraca indica flower. Acta A Mol. Biomol. Spectrosc. 2014, 117, 102-108. http://dx.doi.org/10.1016/j.saa.2013.08.015.
Point 2. How the authors claim that this reaction of MB by Ir-NPs is reduction not adsorption?
Response 2. In addition to the previous explanation, we added more details into the text and other two references (50 and 51) highlighted in yellow. In detail, we added a new figure (Figure S3) in supporting materials to show the reversible leuco-MB oxidation process which demonstrate the reversible MB reduction observed during MB reaction by NaBH4 thus the following sentence was added into the text:
Page 11: “Reduction of MB to leuco-MB is a well-known reversible process promoted by NaBH4 giving a gradual discoloration of the reaction solution and monitored by following the decrease of the peak at 664 nm, but also confirmed from the return at a blue-coloured solution when the final reaction solution is left at the air and the oxygen turns the leuco-MB form to the initial MB one. This behaviour was observed also when the reduction is performed in presence also of Ir-NPs or the extracts. In fact, as can be seen in Figure S3 the increase of the peak at 664 nm is observed when the reaction mixture, after catalytic reaction by NaBH4 in presence of Ir-NP2, is left at the air contact. This result confirms that the reduced form of MB is in the reaction mixture and it is again oxidated to MB by the oxygen.”
Point3. The author should measure the recyclability study of prepared nanoparticles for MB dyes.
Response 3 As noted from the reviewer recyclability in addition to stability studies are important for nanopartcicles catalysts, thus we added into the text some considerations on it, as follow:
Page 14: “Stability and recyclability are important characteristics of a good catalyst [57]. The optimum Ir catalyst synthesized in the present work, Ir-NP2, was stable for a long period of 10 months showing a stable suspension that maintains DLS results and catalytic performance. On the other side, recyclability was not measured because of the small nanoparticles size which hinder their recovery from water. Nevertheless, this preliminary study demonstrated the importance of the extraction conditions to get nanoparticles with optimal catalytic properties for dyes degradation application. Further efforts should be devoted to obtain Ir-NPs in a easily recyclable system such as magnetic bimetallic NPs or by using other supports [57].”
- Ismail, M.; Khan, M.I.; Khan, M.A.; Akhtar, K.; Asiri, A.M.; Khan, S.B. Plant-supported silver nanoparticles: Efficient, economically viable and easily recoverable catalyst for the reduction of organic pollutant. Organomet. Chem. 2019, 33, e4971. https://doi.org/10.1002/aoc.4971.
Compare your results of catalytic reduction with DOI: 10.1002/aoc.4971
Response: We would thank you the reviewer for the suggested reference. We found the reference appropriate for the considerations on recyclability (see point 3), thus we added the above reference in the text as reference 57.
I look forward to hearing from you soon
Best Regards
February 23 2023
Dr. Roberta Del Sole
Corresponding Author:
Dr. Roberta Del Sole
Department of Engineering for Innovation - University of Salento
Via per Monteroni, Km 1, 73100 Lecce , Italy
phone. +39 832 297 256; e-mail: roberta.delsole@unisalento.it
